# Humidity response depends on the small soluble protein Obp59a in *Drosophila*

Jennifer S Sun[1], Nikki K Larter[1,2], J Sebastian Chahda[1], Douglas Rioux[1], Ankita Gumaste[2], John R Carlson[1,2]*

[1]Department of Molecular, Cellular and Developmental Biology, Yale University, New Haven, United States; [2]Interdepartmental Neuroscience Program, Yale University, New Haven, United States

**Abstract** Hygrosensation is an essential sensory modality that is used to find sources of moisture. Hygroreception allows animals to avoid desiccation, an existential threat that is increasing with climate change. Humidity response, however, remains poorly understood. Here we find that humidity-detecting sensilla in the *Drosophila* antenna express and rely on a small protein, Obp59a. Mutants lacking this protein are defective in three hygrosensory behaviors, one operating over seconds, one over minutes, and one over hours. Remarkably, loss of Obp59a and humidity response leads to an increase in desiccation resistance. Obp59a is an exceptionally well-conserved, highly localized, and abundantly expressed member of a large family of secreted proteins. Antennal Obps have long been believed to transport hydrophobic odorants, and a role in hygroreception was unexpected. The results enhance our understanding of hygroreception, Obp function, and desiccation resistance, a process that is critical to insect survival.
DOI: https://doi.org/10.7554/eLife.39249.001

*For correspondence:
john.carlson@yale.edu

**Competing interests:** The authors declare that no competing interests exist.

## Introduction

Hygroreception is a critical sensory modality in the animal world (*Altner and Loftus, 1985*; *Filingeri, 2015*; *Okal et al., 2013*; *Sayeed and Benzer, 1996*; *Shelford, 1918*; *von Arx et al., 2012*). Mosquitoes, for example, use hygroreception to find humans on which to feed, and to find water sources on which to lay eggs (*Okal et al., 2013*; *Takken, 1991*). Hygroreception helps animals avoid desiccation, a peril that is increasing due to climate change. Small insects, which have a high ratio of surface area to volume, are especially vulnerable to water loss (*Gibbs et al., 2003*). The ability to sense humidity levels may allow an insect to avoid dangerously dry conditions or to initiate physiological changes that protect it against desiccation (*Stinziano et al., 2015*).

The antenna functions as a humidity detector in many insects (*Altner and Loftus, 1985*; *Tichy, 1987*; *Yokohari, 1978*). In the antenna of *Drosophila*, humidity detection occurs largely in a three-chambered cavity called the sacculus. The second chamber of the sacculus contains a small number of sensilla that act as hygroreceptors and thermoreceptors (*Enjin et al., 2016*; *Frank et al., 2017*; *Kim and Wang, 2016*; *Knecht et al., 2017*; *Knecht et al., 2016*; *Shanbhag et al., 1995*; *Silbering et al., 2011*). These sensilla belong to a morphological class known as coeloconic sensilla, which are small relative to other sensilla (*Shanbhag et al., 1995*).

The molecular basis of hygroreception remains enigmatic. A major advance was recently made through the discovery that four ionotropic receptors expressed in the sacculus (IR93a, IR25a, IR68a, and IR40a) are required for hygrosensation (*Enjin et al., 2016*; *Frank et al., 2017*; *Kim and Wang, 2016*; *Knecht et al., 2017*; *Knecht et al., 2016*). However, the precise role of these receptors in hygroreception remains unclear. Moreover, the downstream effects of hygrosensory signaling remain poorly understood.

**eLife digest** Some insects have a sense – called hygroreception – that allows them to detect changing levels of moisture in the air. These insects use this sense to avoid becoming too dry, or to find food or places to lay their eggs. In many species, including the fruit fly *Drosophila melanogaster*, the antennae are important for hygroreception. Cells in the antennae produce lots of small proteins called odorant binding proteins, or Obps for short. These proteins are believed mostly to help the antennae to detect various chemical signals in the air, but it was not known if any of these proteins were also involved in hygroreception.

Obp59a is an odorant binding protein that is found in the parts of the antennae that sense moisture, and Sun et al. set out to establish whether it has a role in hygroreception in the fruit fly. A closer look confirmed that Obp59a proteins were indeed found specifically in the moisture-sensitive parts of the antennae, the hygroreceptive sensilla. Further experiments showed that flies without Obp59a could not respond properly to changing humidity over periods of seconds, minutes and hours. These results indicated that Obp59a is important for insect hygroreception.

Perhaps unexpectedly, these mutant flies were also more resistant to drying out. Sun et al. suggest that, because flies without Obp59a struggle with hygroreception, they may also become more cautious to avoid becoming too dry. Further experiments could now test this hypothesis. Since insects like mosquitoes use hygroreception to find their human hosts or choose where to lay their eggs, Obp59a may become a useful target for controlling insect-borne infections. Also, understanding insect hygroreception may yield new insights into how climate change will affect insect populations around the world.

DOI: https://doi.org/10.7554/eLife.39249.002

Also enigmatic has been a family of small secreted proteins called Odorant binding proteins (Obps). These proteins are remarkably numerous, extremely abundant, and highly divergent in sequence (*Graham and Davies, 2002*; *Hekmat-Scafe et al., 2002*; *Menuz et al., 2014*; *Vogt et al., 1989*). There are 52 *Obp* genes in *Drosophila*, of which 27 were found expressed in the antenna in a recent RNAseq analysis (*Larter et al., 2016*; *Menuz et al., 2014*; *Younus et al., 2014*). Five of the 10 most abundantly expressed genes in the antenna are *Obps*. Although Obps are widely believed to carry odorants to odor receptors in olfactory sensilla (*Leal, 2013*; *Leal et al., 2005*), there is limited in vivo evidence to support this role (*Leal, 2013*; *Pelosi et al., 2006*; *Vogt and Riddiford, 1981*), and a recent genetic study found that a mutant olfactory sensillum lacking abundant Obps did not show a decreased magnitude of response to a variety of odorants (*Larter et al., 2016*).

One highly abundant member of the *Obp* family, *Obp59a*, is striking in two respects. First, it is exceptional in its high degree of sequence conservation among insects. Unlike nearly all other *Drosophila Obps*, it has clear orthologs in a variety of insect orders examined (*Vieira and Rozas, 2011*; *Zhou et al., 2010*). Second, it is the most highly localized of the abundant antennal Obps: its expression is restricted to the sacculus (*Larter et al., 2016*).

Here we show that *Obp59a* is expressed in the same sensilla as the *IRs* that are essential to hygroreception. We generate *Obp59a* mutants and find that they are defective in three distinct hygrosensory behavioral paradigms: one operating over the course of seconds, one over minutes, and one over hours. Finally, we show that *Obp59a* mutants survive desiccation better than controls. The results, taken together, add a new dimension to our understanding of hygroreception, of Obp function, and of a process that is critical to insect life and will become even more critical as climate change progresses.

## Results

### *Obp59a* maps to a chamber of the sacculus

We mapped *Obp59a* and examined its specificity of expression, initially by two means. First, detailed in situ hybridization analysis showed that *Obp59a* expression is restricted to the second chamber of the antennal sacculus (*Figure 1A*). Second, we generated an *Obp59a-GAL4* construct, and used it to drive GFP. The expression pattern in the antenna was again restricted to the second

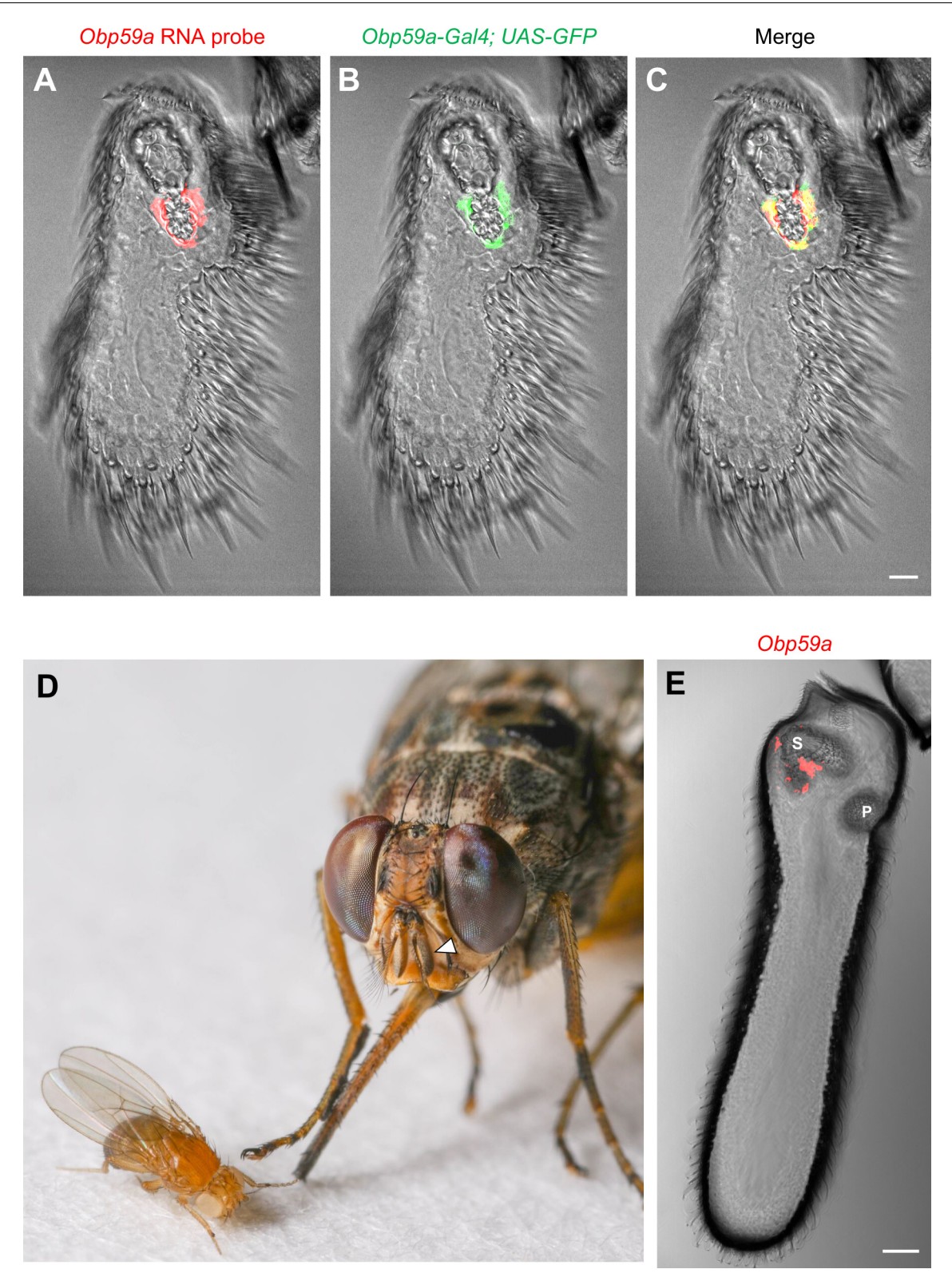

**Figure 1.** *Obp59a* maps to the second chamber of the sacculus. (**A**) In situ hybridization of *Obp59a* to the third segment of the *Drosophila* antenna. Transcript (red) is observed around the second chamber of the sacculus. (**B,C**) *Obp59a-GAL4* drives expression of *UAS-GFP* (green) in the same location as the RNA probe. Scale bar = 12 µm. (**D**) The tsetse fly *Glossina morsitans morsitans* (right); the third antennal segment is indicated by the arrowhead. *Drosophila melanogaster* (left) is shown to illustrate the relative sizes of these flies. Photo courtesy of Dr. Geoffrey Attardo. (**E**) In situ hybridization with

*Figure 1 continued on next page*

*Figure 1 continued*

the *G. morsitans morsitans* ortholog of *Obp59a* to the tsetse antenna shows localization to the sacculus. 'S' indicates the sacculus; 'P' designates a distinct sensory pit observed in the tsetse antenna. Scale bar = 50 μm.

DOI: https://doi.org/10.7554/eLife.39249.003

chamber of the sacculus (*Figure 1B*). The labeling produced by the *Obp59a* probe and by the *Obp59a-GAL4* driver coincided (*Figure 1C*). We found no expression of *Obp59a-GAL4* elsewhere in the fly head or body, or in any of the three larval instars. These results are consistent with data from the Flybase High Throughput Expression Pattern Database, which revealed no expression of *Obp59a* RNA in tissues or developmental stages other than the adult head, which presumably included antennae (*Gelbart and Emmert, 2013*).

We wondered whether the specificity of *Obp59a* expression is conserved in other insects. We examined the antenna of the tsetse fly *Glossina morsitans morsitans,* which diverged from *Drosophila melanogaster* ~75 million years ago (*Wiegmann et al., 2011*) and which carries African sleeping sickness. We found that *Obp59a* again mapped to the sacculus (*Figure 1D,E*).

## *Obp59a* maps to hygrosensitive sensilla

We next asked whether *Obp59a* maps to hygrosensitive sensilla in the second chamber of the sacculus. We carried out a double-label analysis, using an *Obp59a* probe and five *IR-GAL4* constructs that drive expression in the sacculus, four of which label hygrosensitive sensilla in the second chamber and one of which does not.

*IR93a-GAL4* can be seen to label a hygrosensitive neuron in the second chamber (*Figure 2*, top panel of left column, green; the arrowhead indicates the dendrite). *Obp59a* labels cells immediately adjacent to this neuron (*Figure 2*, center and bottom panels, left column). Likewise, *IR25a-GAL4, IR40a-GAL4,* and *IR68a-GAL4* all label hygrosensitive neurons of the second chamber, and in each case, *Obp59a* labels adjacent cells. We did not observe axons or dendrites in any of the cells labeled by *Obp59a*, consistent with its expression in non-neuronal cells of the sensilla, as expected of an *Obp*.

As a negative control, we examined *IR8a-GAL4,* which does not label hygrosensitive neurons of the second chamber. The neurons it labels are not immediately adjacent to cells labeled by *Obp59a* (*Figure 2—figure supplement 1*).

## Obp59a protein localizes within the sensillum shaft

Having shown that *Obp59a* RNA maps to hygrosensitive sensilla in the second chamber of the sacculus, we wanted next to localize Obp59a protein within these sensilla. We generated an anti-Obp59a antibody and found that the antibody labels the second chamber of the sacculus (*Figure 3A*), consistent with that of *Obp59a* RNA (*Figure 3B,C*).

We further validated the antibody by generating an *Obp59a* mutant and asking whether immunolabeling was lost. We deleted the entire *Obp59a* coding region using the CRISPR-Cas9 system (*Supplementary file 1*). When the anti-Obp59a antibody was tested against the antenna of the *Obp59a*[1] deletion mutant, no labeling was observed in the sacculus or anywhere else (*Figure 3D*). These results indicate that the anti-Obp59a antibody specifically labels the Obp59a protein. We note moreover that we examined the structure of the sacculus by confocal microscopy and observed no gross morphological defects in the *Obp59a* mutant (*Figure 3—figure supplement 1*). We cannot exclude the possibility of subtle morphological defects in the sacculus or its sensilla.

We then examined the antibody labeling at higher resolution. We were especially interested in whether the protein was secreted into the shaft of the sensillum (*Figure 3E*), where dendrites reside (*Shanbhag et al., 1999*). In addition to labeling antennal sections (*Figure 3F*) with the antibody (*Figure 3G*), we co-labeled with the *Obp59a* RNA probe to identify the non-neuronal cells that synthesize Obp59a (*Figure 3H*). The merged images show that Obp59a is in fact found within the shafts of the sensilla, as expected of a protein that is secreted by auxiliary cells of a sensillum into the dendritic lymph (*Figure 3I,J*) (*Pelosi and Maida, 1995*). We note that Obp59a contains a signal sequence, consistent with secretion of the protein into the shaft.

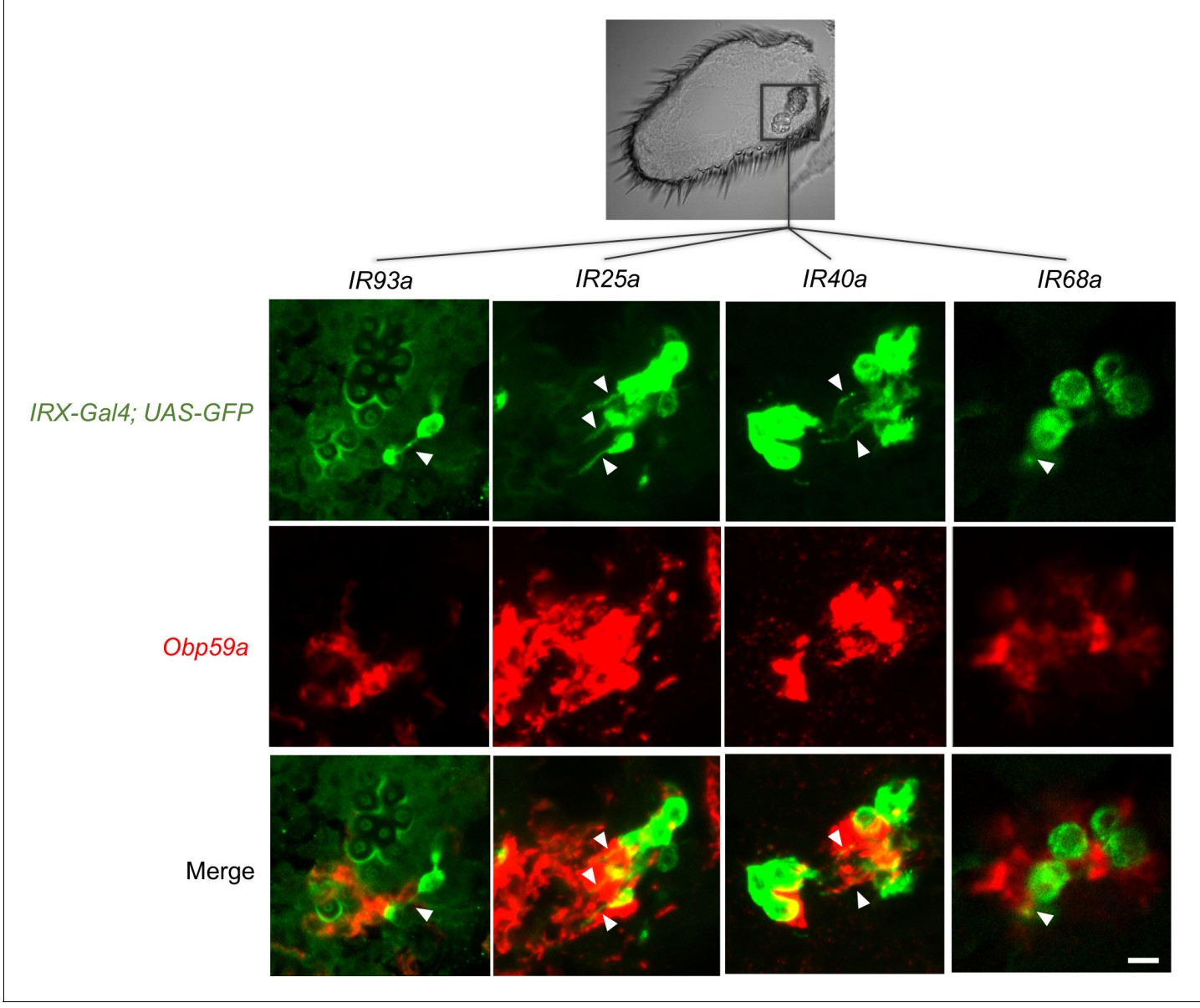

**Figure 2.** *Obp59a* maps to hygrosensitive sensilla. Antennal *IRX-GAL4; UAS-GFP* drivers (green) were used in double-label experiments with an *Obp59a* probe (red). Arrowheads indicate neuronal dendrites. *Obp59a* localizes to cells adjacent to hygrosensitive neurons that express drivers of *IR93a, IR25a, IR40a,* and *IR68a*. The 12 fluorescent images are all from the region of the antenna containing the sacculus (rectangle in black and white image above). The dendrite labeled by the *IR68a-GAL4* driver can be visualized more clearly by examining multiple focal planes. Scale bar of fluorescent images = 6 μm.

DOI: https://doi.org/10.7554/eLife.39249.004

The following figure supplements are available for figure 2:

**Figure supplement 1.** *Obp59a* does not map to sensilla expressing *IR8a-GAL4*.
DOI: https://doi.org/10.7554/eLife.39249.005
**Figure supplement 2.** Hygrosensitive neurons are present in *Obp59a[1]*.
DOI: https://doi.org/10.7554/eLife.39249.006

## An *Obp59a* mutant is defective in behavioral response to humidity

We asked whether *Obp59a* is required for response to humidity in three different behavioral paradigms. Prior to testing, mutants carrying the *Obp59a* mutation were backcrossed to the control stock for five generations to minimize genetic background effects.

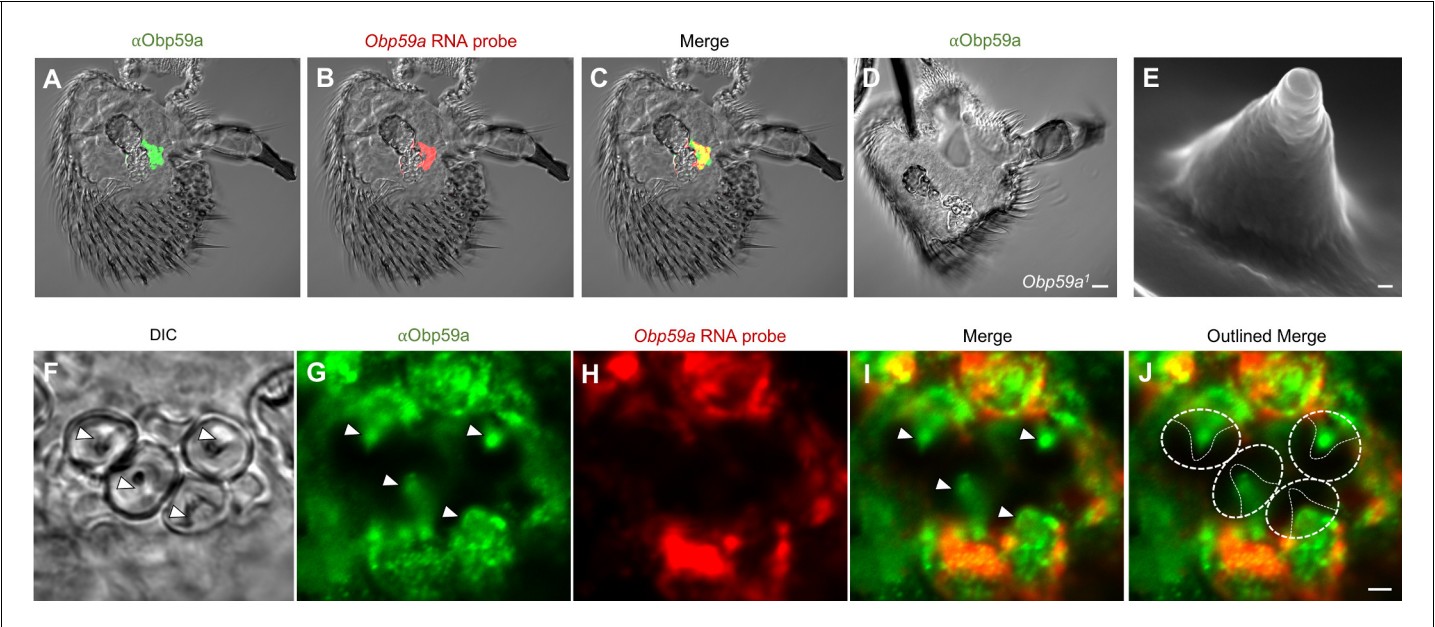

**Figure 3.** Obp59a protein localizes within the sensillum shaft. (A) αOBP59a antibody labels the second chamber of the sacculus, in a pattern that overlaps with that of the *Obp59a* RNA probe (B,C). (D) The αOBP59a antibody does not label *Obp59a[1]*. Scale bar = 18 μm. (E) Ultrastructure of a hygrosensory sensillum in the second chamber of the sacculus. Scale bar = 100 nm. (F) Antennal section through the second chamber, showing hygrosensitive sensilla (arrowheads). (G–J) Double-labeling with αOBP59a antibody and the *Obp59a* RNA probe, showing that Obp59a protein is localized within the shafts of hygrosensitive sensilla, which in (J) are outlined by thin solid lines within the dotted regions of interest; immunolabeling is also observed in non-neuronal cells that express the *Obp59a* RNA, presumably representing nascent protein before secretion. Scale bar = 2 μm.
DOI: https://doi.org/10.7554/eLife.39249.007

The following figure supplement is available for figure 3:

**Figure supplement 1.** Gross morphology of the sacculus is normal in *Obp59a[1]*.
DOI: https://doi.org/10.7554/eLife.39249.008

First we tested hygrotaxis in a paradigm that operates on a time scale of minutes (*Ji and Zhu, 2015*). Flies are placed in a Petri dish that contains an inner region held at high humidity and an outer region held at low humidity (*Figure 4A*). Humidity was controlled through the use of saturated salt solutions and was verified with a hygrometer (*Enjin et al., 2016*). Initially, flies were distributed uniformly on the plate, and their distributions were then measured at 10 s intervals over the course of 5 min. A hygrotaxis index was calculated at each time point as the fraction of flies in the central region of high humidity, following established convention (*Ji and Zhu, 2015*). Since the area of high humidity was ~1/10 that of the total area, and the distribution of the flies was initially uniform, the initial index was ~0.1.

When the inner region was at 70% relative humidity (RH) and the outer region was at 20% RH, the control flies showed a rapid hygrotaxis behavior: control flies quickly began to move inward toward the more humid region (*Figure 4B*). By contrast, the distribution of *Obp59a* mutants showed little if any change. A striking phenotype was also observed when the inner and outer RH were 96% and 20%, respectively (*Figure 4C*), and 96% v. 70% (*Figure 4D*). When inner and outer RH were equal (and set to 45%), neither genotype showed any place preference (*Figure 4E*). We note that the hygrotaxis responses were very rapid in these experiments, which provides an explanation for why the values at the first time points, taken ~15 s after the flies were distributed uniformly in the dish, were generally somewhat greater than 0.1.

We carried out a rescue experiment to determine whether the phenotype in fact mapped to the *Obp59a* gene. We found that when inner and outer RH were 70% and 20%, mutant flies carrying both an *Obp59a-GAL4* construct and a *UAS-Obp59a* construct showed stronger responses than mutant flies carrying either construct alone (*Figure 4F*). Rescue was also observed in the 70% v. 20% and 96% v. 20% cases (*Figure 4G,H*). All genotypes showed no place preference when inner and outer RH were both set to 45% (*Figure 4I*).

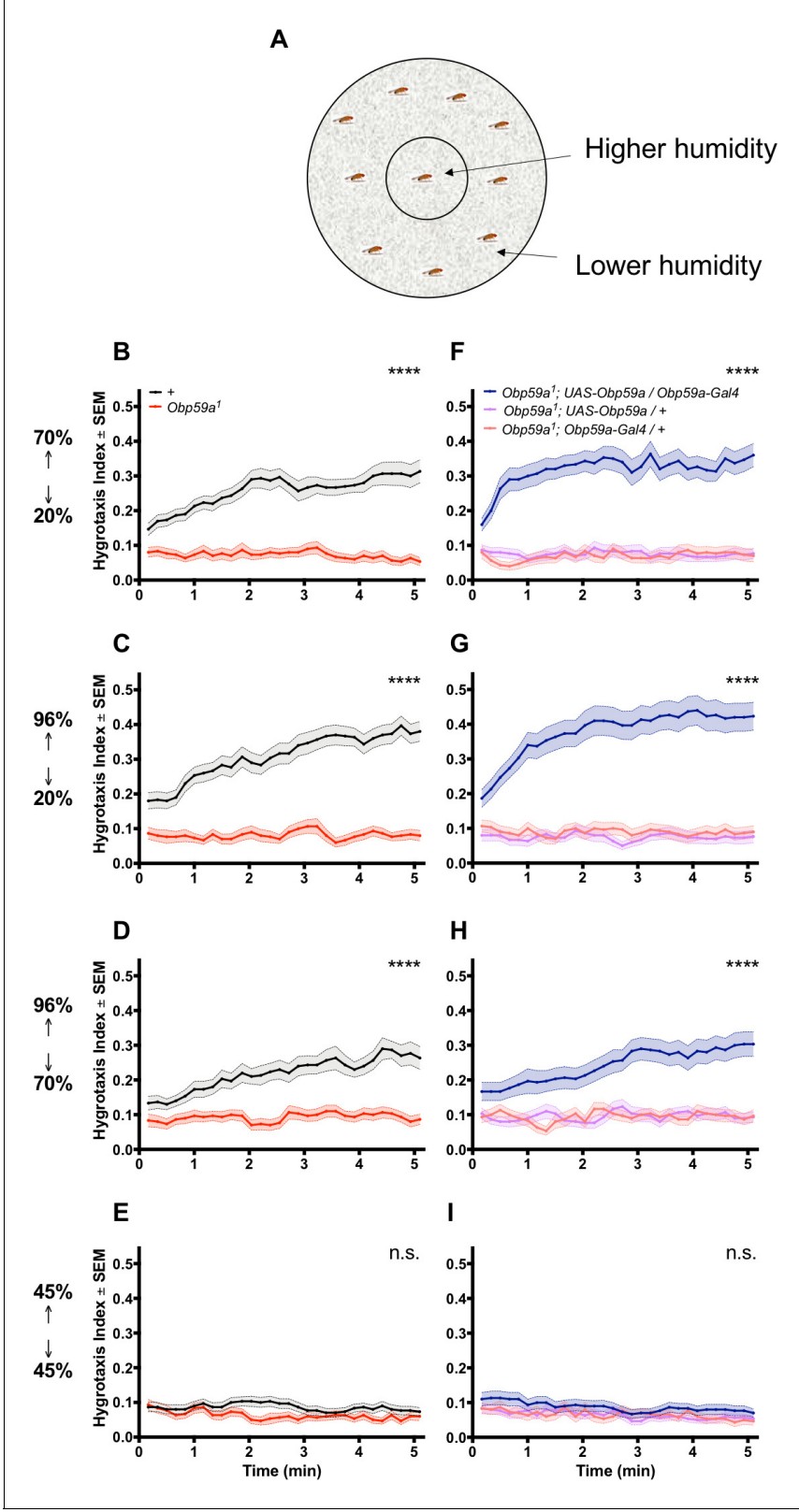

**Figure 4.** An *Obp59a* mutant is defective in a fast hygrotaxis behavior. (**A**) Hygrotaxis paradigm in a Petri dish. (**B–D**) Hygrotaxis response of *Obp59a[1]* (red) is lower than control (black) in all cases in which flies are given a choice between two different humidities. (**E**) Neither genotype shows a response when given a choice between two identical humidities. (**F–H**) Expression of a *UAS-Obp59a* rescue construct (dark blue) increases humidity response

*Figure 4 continued on next page*

*Figure 4 continued*

above each parental control value. (I) Under the condition of identical humidities, no genotype shows a response. ****p<0.0001, n.s. = not significant, Friedman test with Dunn's multiple comparisons test, n = 30 replicates.

DOI: https://doi.org/10.7554/eLife.39249.009

The following figure supplements are available for figure 4:

**Figure supplement 1.** Desiccated *Obp59a*[1] flies are defective in a fast hygrotaxis behavior.

DOI: https://doi.org/10.7554/eLife.39249.010

**Figure supplement 2.** The hygrotaxis phenotype maps to *Obp59a*.

DOI: https://doi.org/10.7554/eLife.39249.011

Similar hygrotaxis phenotypes were observed when flies were desiccated prior to the test (*Figure 4—figure supplement 1A–D*). We note that the desiccated *Obp59a* mutants appeared to gravitate toward the more humid region near the end of the test period (*e.g. Figure 4—figure supplement 1A*). This response may arise from extreme thirst and perhaps other pathways for humidity detection (*Liu et al., 2007*).

We note that the behavior of control flies that were desiccated was not dramatically different from those that were not (*Figure 4* vs. *Figure 4—figure supplement 1*). An earlier study showed using a different paradigm that hygrotaxis behavior can be altered by desiccation (*Knecht et al., 2017*); perhaps we have not observed a major alteration because of the different geometry, larger size, and shorter duration of the paradigm shown in *Figure 4*, or because of differences in the desiccation procedure we used, which for example did not provide a sucrose source to flies.

As further confirmation that the *Obp59a* gene is required for normal hygrotaxis behavior, we tested an independent allele, *Obp59a*[2], which was also backcrossed five times to the control strain. We again found a strong defect in all three cases: 70% v. 20%, 96% v. 20%, and 96% v. 70% RHs (*Figure 4—figure supplement 2A–D*). As another control, we tested a mutant of a related gene, *Obp28a*, which was generated in the same manner as the *Obp59a* mutants, and found no defects in any test in this paradigm (*Figure 4—figure supplement 2E–H*). We further tested *Obp59a*[2] and *Obp28a* following desiccation and again found a phenotype for *Obp59a*[2] but not *Obp28a* (*Figure 4—figure supplement 2I–P*). These results provide additional evidence that the humidity response phenotype maps to the *Obp59a* gene.

Next we tested *Obp59a*[1] in a second paradigm in which we measured humidity preference over the course of hours (*Figure 5A*) (*Knecht et al., 2016*). Flies were given a binary choice between higher and lower regions of humidity, and a 'wet preference' was calculated at one hour intervals as (H-L)/(H + L), where H is the number of flies in the region of high humidity and L is the number in the region of low humidity. Thus, the wet preference may vary between 1.0 (complete preference for high humidity) to −1.0 (complete preference for low humidity). We first gave flies a choice between 70% and 20% RH and found that by the first time point (5 min), control flies showed a strong preference for high humidity; this preference continued for 24 hr (*Figure 5B*). The response of *Obp59a*[1] was much lower at this first time point, and remained lower over the course of many hours. Response of the mutant was also lower when given a choice between 70% and 50% RH, and between 96% and 70% RH (*Figure 5C,D*). The *Obp59a*[1] mutant retains some limited ability to respond to humidity, as evidenced most clearly by the responses in the first two cases at 24 hr, when flies may be very thirsty (*Figure 5B,C*). We note finally that although the preference we observe in control flies between 96% and 70% RH is weak, the valence is opposite that found in some other studies (*Enjin et al., 2016*; *Knecht et al., 2017*; *Perttunen and Salmi, 1956*). *Knecht et al., 2017* and *Perttunen and Salmi, 1956* have shown that the valence of this preference is dependent on the hydration status of the flies, and it is possible that the different preference we have observed in our experiments reflects a difference in the conditions in which the flies are cultured.

We then tested the response of *Obp59a*[1] to humidity in a third paradigm: a modified proboscis extension response (PER) test (*Figure 5E*) (*Ji and Zhu, 2015*). We measured the response of a desiccated fly to the water vapor emanating from a moistened cotton swab. The response of *Obp59a*[1] was lower than that of the control genotype (*Figure 5F*). Neither genotype responded to a dry stimulus.

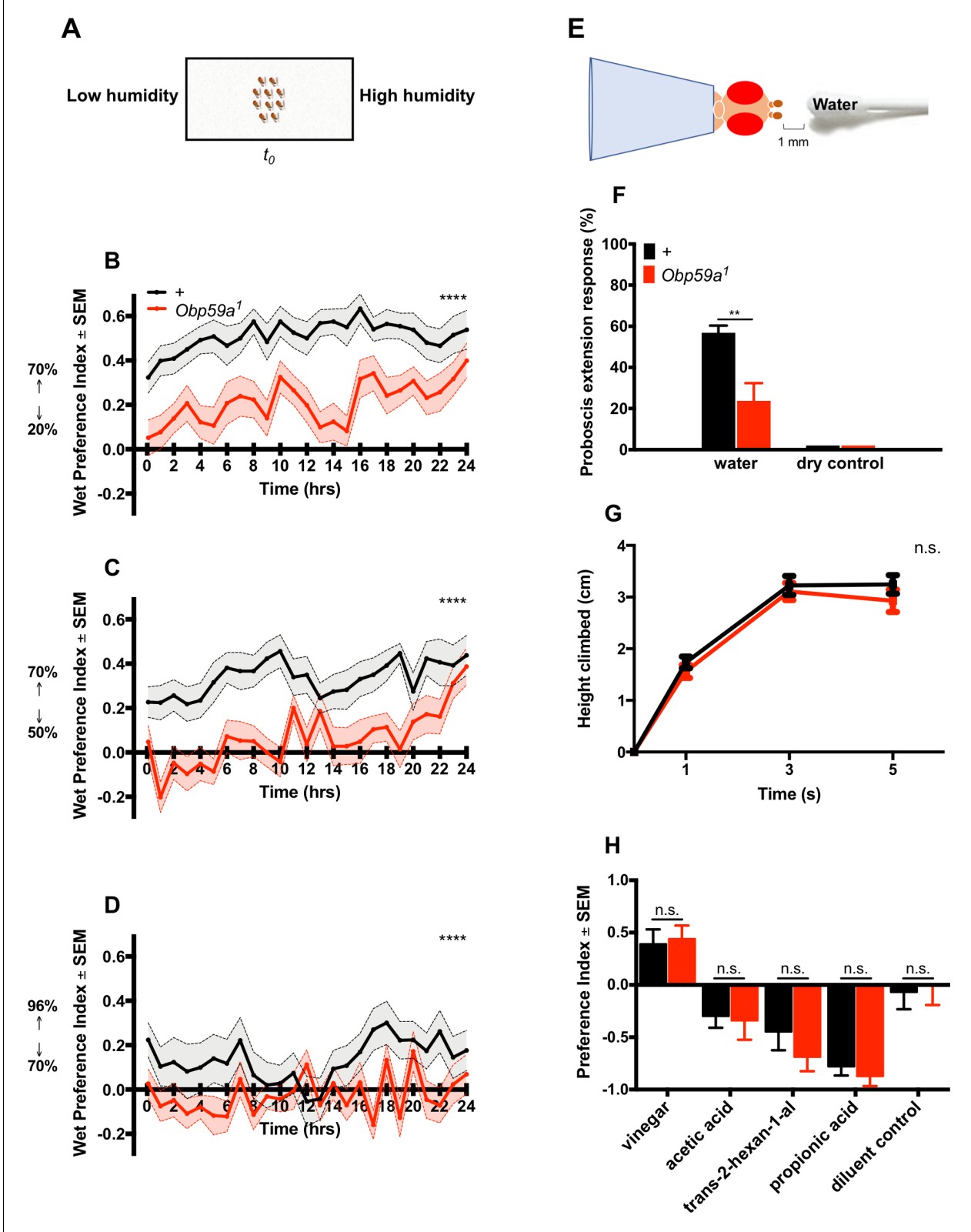

**Figure 5.** *Obp59a* is defective in humidity preference, but not in other behaviors. (**A**) A humidity preference paradigm based on that of *Knecht et al. (2016)*. (**B–D**) Control flies (black) show a sustained preference for higher humidities; *Obp59a[1]* flies show reduced responses during most of the 24 hr test period. We note that the first time point was taken at 5 min, by which time a number of flies had migrated to the region of high humidity. ****p<0.0001, Friedman test with Dunn's multiple comparisons test, n = 24 replicates. (**E**) Modified Proboscis Extension Response paradigm. (**F**)

*Figure 5 continued on next page*

*Figure 5 continued*

Mutants have reduced PER responses to water vapor. **p<0.01, Mann-Whitney test, n = 22 replicates. (G) RING test of climbing behavior. n.s. = not significant, Mann-Whitney test, n = 10 replicates. (H) Mutants show no phenotype in an olfactory trap assay. Apple cider vinegar was diluted $10^{-3}$, and trans-2-hexenal was diluted $10^{-2}$, both in paraffin oil. n.s. = not significant, Mann-Whitney test, n = 15 replicates.
DOI: https://doi.org/10.7554/eLife.39249.012

Could *Obp59a* mutants be defective in these paradigms because they have lost the humidity-sensing neurons of the sacculus? We addressed this question with three independent reagents: an anti-IR93a antibody, an anti-IR25a antibody, and an *IR68a-GAL4* driver. All three reagents labeled neurons in the second chamber of the sacculus of *Obp59a[1]*, in a pattern comparable to that observed in controls (*Figure 2—figure supplement 2*).

Do *Obp59a[1]* mutants show defects in these three behavioral paradigms because of a general deterioration in health or mobility? We tested the ability of *Obp59a[1]* flies to rapidly climb walls in the Rapid Iterative Negative Geotaxis (RING) assay and found that *Obp59a[1]* showed the same robust climbing behavior as wild type (*Figure 5G*). *Obp59a[1]* also shows robust chemosensory responses to an attractive stimulus and several repellent stimuli in an olfactory paradigm (*Figure 5H*). The repellent stimuli include acetic acid and propionic acid, which are detected at least in part by sensilla in the third chamber of the sacculus (*Ai et al., 2013*).

## Loss of *Obp59a* increases desiccation resistance

If *Obp59a* mutants do not perceive humidity normally, are there consequences for their physiology? Specifically, we wondered whether mutants might be affected in desiccation resistance. Desiccation is a critical threat to insect survival, and a major function of the humidity detection system is likely to prevent flies from dying of desiccation. Accordingly, we placed *Obp59a[1]* males in a chamber under desiccating conditions (0% RH) and measured survival.

Remarkably, *Obp59a[1]* males survived longer than control males (*Figure 6A*, p<0.0001). Consistent with these results, we found in an independent experiment that *Obp59a[2]* males also survived longer than control males (*Figure 6B*, p<0.0001). Females survived longer as well, in the case of both *Obp59a[1]* and *Obp59a[2]* (*Figure 6C,D*, p<0.0001 in both cases).

Survival of males and females of both alleles was normal at 70% RH (*Figure 6—figure supplement 1A–D*). As another control, *Obp28a* mutants showed normal survival, in the case of both males and females and at both 0% and 70% RH (*Figure 6—figure supplement 1E–H*). One interpretation of these results is that the loss of Obp59a leads to an abnormal pattern of humidity signaling, which triggers defensive physiological changes in the fly that protect it from desiccation.

## Discussion

We have identified a component essential for normal hygroreception, a critical process about which remarkably little is known. We found that Obp59a is localized within hygrosensory sensilla, and that loss of Obp59a reduced the response of the animal to humidity in three distinct behavioral paradigms. Animals lacking Obp59a displayed a major increase in resistance to desiccation. We were surprised to find such a profound role in humidity response for a member of the Obp family.

## A role for an Obp in humidity detection

Expression of *Obp59a* in *Drosophila* is highly localized to the sacculus. The tsetse ortholog is also expressed in the sacculus, suggesting that its location has been conserved for at least 75 million years. Within the sacculus of *Drosophila* the protein is found in the shafts of hygrosensory sensilla, where dendrites of hygrosensory neurons are located (*Shanbhag et al., 1999*).

Expression levels of *Obp59a* are remarkably high. *Obps* are the most abundantly expressed genes in the antenna: in a recent RNAseq analysis, *Obp19d* and *Obp83a* were detected at 31,000 and 25,000 RPKM (reads per kilobase per million mapped reads) respectively, while a typical *Odor receptor (Or)* gene was expressed at ~40 RPKM (*Menuz et al., 2014*). *Obp59a* was expressed at ~2,000 RPKM, and showed the most highly restricted expression of the antennal *Obp* genes (*Larter et al., 2016*): most other *Obps* are expressed in many more sensilla. These results suggest

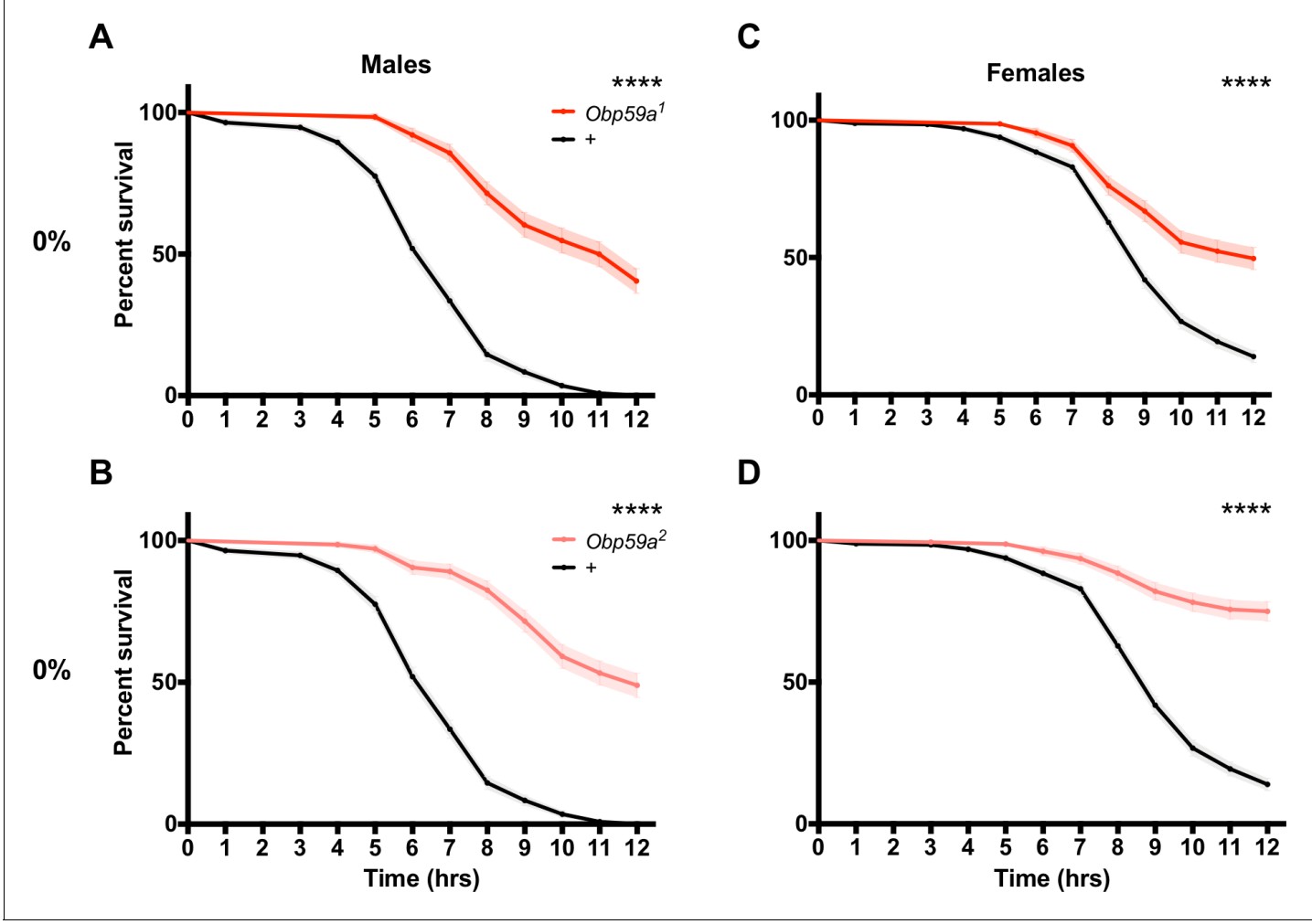

**Figure 6.** *Obp59a* mutations confer increased desiccation resistance. Mutant flies survive longer under desiccating conditions than controls. (**A**), *Obp59a^1* males. (**B**), *Obp59a^2* males. (**C**), *Obp59a^1* females. (**D**), *Obp59a^2* females. The control data in (**A**) and (**B**) are the same because the two mutant alleles were tested in parallel; likewise for (**C**) and (**D**). ****p<0.0001, log-rank (Mantel-Cox) test, n = 20–28 replicates. Supplemental Information Legends.

DOI: https://doi.org/10.7554/eLife.39249.013

The following figure supplement is available for figure 6:

**Figure supplement 1.** Normal survival of *Obp59a* at 70% humidity and normal survival of *Obp28a*.

DOI: https://doi.org/10.7554/eLife.39249.014

that the level of *Obp59a* expression in an individual hygrosensitive sensillum is comparable to that of the most abundantly expressed *Obps*.

Not only is the expression pattern of *Obp59a* conserved, but its sequence is conserved as well (*Stanley and Kulathinal, 2016*; *Vieira and Rozas, 2011*; *Zhou et al., 2010*). Obp59a is one of only two Obps with clear orthologs across insect orders. This conservation suggests that its structure represents a good solution to a difficult problem that is common to many insects.

A role for an Obp in humidity detection was unexpected. Antennal Obps are widely believed to transport hydrophobic odorants through the aqueous sensillum lymph to odor receptors in the dendritic membranes of olfactory receptor neurons (*Leal, 2013*). An *Obp59a* mutant was normal in response to both attractant and repellent odorants in behavioral tests. Obp59a seems unlikely to carry water molecules across the aqueous lymph to the dendrites of hygrosensory neurons.

One proposed model for the mechanism of hygroreception is that a change in humidity alters the structure of hygrosensory sensilla, with the structural change being transduced into neuronal

responses (*Altner and Loftus, 1985*). It is conceivable that *Obp59a* mutants contain a subtle defect in the structure or composition of the sensillum, and therefore do not undergo a normal structural change in response to changes in humidity. Such a role might fit well with the high abundance of Obp59a within hygrosensory sensilla. Obp59a could possibly affect sensillum structure or composition via a role in transporting hydrophobic components of the cuticular wall of the sensillum.

The concept of an alternative role for an antennal Obp is consistent with a recent study showing that the classic odorant-transport model may not apply to all antennal Obps and all olfactory sensilla (*Larter et al., 2016*). An Obp-to-sensillum map was constructed for all 10 of the abundant Obps, and when one particular sensillum, ab8, was genetically depleted of its sole abundant Obp, it showed a robust electrophysiological response to odorants; in fact the peak response was increased in many cases. These results suggested that Obp28a is not required for the transport of odorants to Ors in the ab8 sensillum.

Taken together, the results reveal an unexpected molecular component required for normal humidity response. Our findings add further support to the concept that antennal Obps do not have a single, unifying function, but rather play diverse roles. The results also identify a new target that could be useful in controlling insect vectors that rely on humidity to find their human hosts and oviposition sites.

## The role of humidity detection in fitness

Deletion of a gene in *Drosophila* often causes a decrease in fitness. We were surprised that deletion of *Obp59a* caused an increase in fitness under desiccating conditions.

How might greater desiccation resistance be achieved? We suspect that a constellation of metabolic changes together produce desiccation resistance. How would these changes be triggered? We speculate that owing to the *Obp59a* defect, the pattern of integrated sensory input from all humidity-sensing circuits in the fly – those activated by the neurons studied here and others (*Yao et al., 2005*), including other antennal neurons expressing certain Trp channels (*Liu et al., 2007*) – is abnormal. This abnormality would trigger the induction of a defensive state that protects the fly against the existential threat of desiccation.

# Materials and methods

**Key resources table**

| Reagent type (species) or resource | Designation | Source or reference | Identifiers | Additional information |
|---|---|---|---|---|
| Gene (*D. melanogaster*) | *Obp59a* | NA | FLYB:FBgn0034766 | |
| Gene (*G. morsitans morsitans*) | *Obp59a* | NA | Vectorbase: GMOY006081 | |
| Genetic reagent (*D. melanogaster*) | *wCS* | DOI: 10.1016/j.neuron.2014.07.012 | | |
| Genetic reagent (*D. melanogaster*) | *Obp28a⁻* | DOI: 10.7554/eLife.20242 | | |
| Genetic reagent (*D. melanogaster*) | *Obp59a¹ and Obp59a²* | this paper | | |
| Genetic reagent (*D. melanogaster*) | *IR40a-GAL4* | Bloomington Drosophila Stock Center | BDSC:41727 | |
| Genetic reagent (*D. melanogaster*) | *IR25a-GAL4* | Bloomington Drosophila Stock Center | BDSC:41728 | |

*Continued on next page*

*Continued*

| Reagent type (species) or resource | Designation | Source or reference | Identifiers | Additional information |
|---|---|---|---|---|
| Genetic reagent (*D. melanogaster*) | *IR93a-GAL4* | DOI: 10.1016/j.cub.2016.03.049 | FLYB:FBtp0102896 | |
| Genetic reagent (*D. melanogaster*) | *IR68a-GAL4* | DOI: 10.7554/eLife.26654 | | |
| Genetic reagent (*D. melanogaster*) | *IR8a-GAL4* | Bloomington Drosophila Stock Center | BDSC:41731 | |
| Genetic reagent (*D. melanogaster*) | *Obp59a-GAL4* | this paper | | Created using 5' and 3' fragments cloned into pBGRY1. Injected into *attP2* site. |
| Genetic reagent (*D. melanogaster*) | *attP2* | Bloomington Drosophila Stock Center | BDSC:8622 | |
| Genetic reagent (*D. melanogaster*) | *IR8a-GAL4* | Bloomington Drosophila Stock Center | BDSC:51324 | |
| Genetic reagent (*D. melanogaster*) | *UAS-GFP* | DOI: 10.1016/S0896-6273 (00)80701–1 | | |
| Protein | Obp59a | this paper | Novoprotein:SBP5A | Created using insect cell-optimized cDNA cloned into pFastbac1. |
| Antibody | mouse αObp59a | this paper | | 1:250, mouse polyclonal. |
| Antibody | rabbit αIR93a | DOI: 10.7554/eLife.17879 | | 1:250 |
| Antibody | guinea pig αIR25a | DOI: 10.7554/eLife.17879 | | 1:250 |
| Antibody | goat αmouse Alexa 488 | ThermoFisher | ThermoFisher:A-32723 | 1:500 |
| Antibody | goat αrabbit Alexa 488 | ThermoFisher | ThermoFisher:A-11034 | 1:500 |
| Antibody | goat αguinea pig Alexa 488 | ThermoFisher | ThermoFisher:A-11073 | 1:500 |
| Recombinant DNA reagent | pU6-BbsI-chiRNA (plasmid) | Addgene | Addgene:45946 | pBS-SK(+) vector backbone |
| Recombinant DNA reagent | pHD-DsRed-attP (plasmid) | Addgene | Addgene:51019 | pJ204 vector backbone |
| Recombinant DNA reagent | pBGRY1 (plasmid) | GenBank | GenBank:KM016698 | |

## Fly strains

Flies were reared on standard cornmeal-dextrose agar food at 25°C and 60% humidity. Flies used in behavioral experiments were backcrossed to *wCS* for at least five generations to minimize genetic background effects.

The following *IR-GAL4* lines were used: *IR40a-GAL4* (BDSC #41727), *IR25a-GAL4* (BDSC #41728), *IR93a-GAL4* (from Dr. Marco Gallio), *IR68a-GAL4* (from Dr. Paul Garrity), *IR8a-GAL4* (BDSC #41731).

*Obp59a-GAL4* flies were created using 5' and 3' fragments cloned into pBGRY1. Primers used for the 5' end were: CTGCTGTTTGATGGCTTGC (−500 to −480) and CTTGGGAACTGAATGGAGGA (−1 to −20, reverse complement). Primers for the 3' end were: GATTAAACTCACCCCACTTTTTAGG (+1 to+25) and AACATTTTAATCAGAAACTAAATACACAGCT (+469 to+500, reverse complement). 40 bp of yeast sequence remained between the GAL4 stop and the attB3 site, into which the 3' end of *Obp59a* is cloned. Plasmids were injected into *y¹w⁶⁷ᶜ²³; P{CaryPattP2* (BDSC #8622) flies.

### *Obp59a* deletion

Guide chiRNAs were cloned into *pU6-BbsI-chiRNA* plasmids. Selected cut sites were located 1-nt upstream of the 5' end, and 162-nt upstream of the 3' end, removing 80.6% of the coding sequence of *Obp59a*. Homology arms extending 1.02 kb upstream and 1.02 kb downstream of the cut site were incorporated into the *pHD-DsRed-attP* vector (*Gratz et al., 2014*). Cloning methodology was described in *Larter et al. (2016)*. *w[1118]; Pbac{y[+mDint2]=vas-Cas9}VK00027* (BDSC #51324, [*Gratz et al., 2013*]) embryos were injected by Bestgene, Inc. (Chino Hills, CA). Non-sibling G1 adults expressing *DsRed* were identified. Primer pairs extending beyond the *Obp59a* coding sequence and homology arms were used to verify gene deletion. Mutation strategy is provided in *Supplementary file 1*.

### Expression analysis

7 day old flies were immobilized with $CO_2$. Four female and four male flies were placed in a collar fashioned from two inward-facing razor blades stabilized on a stack of microscope slides, as described previously (*Larter et al., 2016*). Animals were covered in OCT compound (Tissue-Tek) and rapidly frozen on dry ice. 14 μm sections were collected on slides and stored at −80°C until use.

Double-label in situ hybridization utilized *Obp59a* RNA probes and *GAL4* lines driving expression of *UAS-mCD8-GFP*. RNA probes were synthesized as described in *Larter et al. (2016)*. The staining protocol was previously described (*Menuz et al., 2014*).

*Glossina* experiments were similarly conducted using *GmObp59a* RNA probe, with some modifications. A 450 bp segment of *GmmObp59a* was PCR-amplified from *Glossina morsitans morsitans* antennal cDNA using the Forward Primer: TGCCGTACAGATGATGGACC, and Reverse Primer: GGCGATGCTGTGATTCCAAG. From this *GmmObp59a* DNA template, an unfragmented digoxigenin (DIG)-labeled RNA antisense probe was synthesized using standard methods. Antennae were cryosectioned at 40 μm, RNA probes were hybridized at 55°C overnight, sheep anti-DIG-POD primary antibodies were incubated for 45 min, and Cy3 TSA was used for signal detection.

Immunohistochemistry was conducted using polyclonal mouse αObp59a produced by Cocalico Biologicals, Inc. (Stevens, PA) from insect cell-expressed Obp59a prepared by Novoprotein (Summit, NJ), and anti-IR93a and anti-IR25a antibodies provided by Richard Benton.

### Ultrastructure analysis

7 day old female flies were collected and prepared in collars as described above for expression analysis. Antennae were sectioned at 10 μm and sections mounted on 9.5 mm aluminum stubs (Electron Microscopy Services #75180) using carbon paint (Electron Microscopy Services #12691–30). Samples were coated with 8 nm iridium with a Cressington 208 iridium sputtering tool. SEM was carried out with a Hitachi SU-70 electron microscope equipped with solid-state backscatter detector for enhanced imaging of grain boundaries.

### Behavioral assays

#### Preparation of animals

The population density of flies was carefully controlled – 7 female and two male flies were introduced in bottles and allowed to mate and lay eggs. After four days, the adults were cleared.

Behavioral experiments utilized 7 day old flies immobilized on ice and sorted by sex, unless otherwise specified. For behavioral testing of desiccated flies, flies were first placed in empty 29 mm food vials in the dark on 200 mL desiccant (W.A. Hammond Drierite Co., Ltd.) in a sealed 1000 mL Nalgene jar (ThermoFisher) for 5 hr at 25°C. No food source was present during this desiccation period. For behavioral testing of starved flies, flies were first starved in the dark at 70% RH at 25°C for 24 hr in the presence of filter paper that had been moistened with water.

#### Petri dish hygrotaxis arena

The assay was adapted from *Ji and Zhu (2015)* (*Figure 4A*). Ten flies, five male and five female, were gently placed in 100 × 15 mm Petri dishes (BD Falcon) and sealed with size 66–68 nylon mesh (FabricandSewing.com). Salt solutions influencing humidity were prepared in a separate Petri dish. 35 × 10 mm Petri dishes (BD Falcon) were glued in the center of 100 × 15 mm dishes. High humidity above the central dish was created with 10 ml $ddH_2O$ (96% RH) or highly concentrated NaCl (70%

RH). Lower humidity above the outer dish was created with 20 ml concentrated LiCl (20% RH) or NaCl (70% RH). The 100 × 15 mm Petri dish was then covered and allowed to equilibrate for two hours in an environmental room maintained at a constant temperature of 25°C and 70% humidity; the assays were performed under these conditions as well. The RH in the region above each solution was stable and the boundary between them was sharp, as confirmed with a hygrometer (Hanna Instruments). Petri dishes containing flies were then inverted atop the Petri dishes containing solutions, so that flies were free to walk on the nylon mesh. Flies were recorded under infrared lighting for 300 s. The fraction of flies in the central dish was scored blind to genotype at 5 s intervals. Hygrotaxis index = (# flies in higher humidity) / (total # of flies).

## Multidish humidity preference arena

The assay was adapted from *Enjin et al. (2016)* (*Figure 5A*). Nunclon Δ Multidishes (ThermoFisher #167063) were prepared with a different solution in each half: either saturated LiCl (20% RH), $MgCl_2$ (50% RH), or NaCl (70% RH), or $ddH_2O$ (96% RH). Mesh was stretched over the Multidish and sealed with the edges of a lid whose face had been removed. Above this frame a second, intact lid was placed. This second lid contained on its inner surface two 5 mm x 22 mm x 60 mm rectangular arenas constructed from trimmed Hybrislip covers (Sigma-Aldrich #GBL716022). Each half of the length of each arena was over a different solution. Each arena housed a different population of flies, mutant in one arena and control in the other. In this way, the two genotypes could be tested in parallel in the same chamber. Before adding the flies, the Multidish was allowed to equilibrate for two hours in the environmental room at 25°C and 70% RH as described above. Then 20 flies, half male and half female, which were not desiccated or starved beforehand, were gently tapped in through holes in the cover so that flies walked on the nylon mesh within each arena. The holes were then resealed with clear tape. Flies were recorded under infrared lighting for 24 hr. A wet preference index (WPI) of flies was scored blind to genotype at 1 hr intervals: (# flies in higher humidity - # flies in lower humidity) / (total # of flies). The tests were performed at 25°C and 70% RH.

## Proboscis extension response

This assay was conducted as described in *Ji and Zhu (2015)* (*Figure 5E*). A desiccated fly was immobilized in a cut pipette tip with proboscis exposed. A dry cotton swab was held at 1 mm distance from the antennae. Water was then presented on a cotton swab held at 1 mm distance from the antennae. A fly was recorded as a responder to the dry or wet cotton swab if it extended its proboscis twice within one minute. 1 M sucrose was then presented to the fly on a cotton swab, but this time by directly touching the proboscis. Data were discarded for the fly if it did not extend its proboscis to sucrose (~25% of the flies). Proboscis extension response = (# flies responding to water) / (total # of flies) x 100%.

## Climbing assay

The assay was adapted from *Nichols et al. (2012)*; *Koh et al. (2014)*. 10 newly emerged flies were anesthetized by $CO_2$ and placed in individual vials with 1 cm of fly food. Flies recovered from anesthetization for seven days in an incubator at 25°C and 70% RH. The climbing assay was conducted under red light (Bright Lab #35010). A camera was used to record climbing behavior. Flies were gently tapped into empty vials and knocked to the bottom. The percentage of flies at each centimeter interval was scored blind to genotype after 1, 2, and 3 s.

## Two choice TRAP assay

TRAP assays were designed as described in *Woodard et al. (1989)*. A 25 µl drop of odorant diluted in paraffin oil, or a drop of paraffin oil alone, were placed on a filter disc at each trap's entrance. Two traps were placed side-by-side in 100 × 20 mm Petri dishes, one containing odorant and the other the control. 20 starved flies, half male and half female, were placed in the Petri dish. Flies were given 24 hr to select a trap to enter. The number of flies in each trap were counted, and a preference index was calculated: (# flies in stimulus trap - # flies in control trap) / (total # of flies in traps).

## Desiccation survival

A survival assay was modified from protocols established in the lab of Dr. Mimi Shirasu-Hiza. Two days after eclosion, flies were distributed to new bottles and allowed to mate for 24 hr. This two-day collection window was selected to ensure that flies were approximately the same age. Flies were then sorted by sex over $CO_2$. 20 sex-segregated flies were taken from culture bottles and placed in an empty 35 × 10 mm Petri dish. Dishes were then placed in 150 mm diameter x 15 mm chambers with different humidities: 0% RH (produced using 25 mL Drierite) or 70% RH (25 mL saturated NaCl), at 25℃ in the dark. No food source was present during the desiccation period. Numbers of dead flies were scored blind to genotype over evenly-spaced time intervals. Survival curves were generated using Prism.

## Acknowledgements

We thank Dr. Melissa Harrison, Dr. Kate O'Connor-Giles, and Dr. Jill Wildonger for plasmids pDsRed-attP (Addgene plasmid 51019) and pU6-BbsI-chiRNA (Addgene plasmid 45946 [*Gratz et al., 2013*]); Dr. Emanuela Zaharieva and Dr. Marco Gallio for *IR93a-GAL4*; Zachary Knecht, Joyner Cruz, and Dr. Paul Garrity for *IR68a-GAL4*. We thank Richard Benton for anti-IR93a and anti-IR25a antibodies. We are grateful to Dr. Mimi Shirasu-Hiza for advice with the desiccation survival assay, Dr. Geoffrey Attardo for the photo in *Figure 1D*, Brian Weiss and Serap Aksoy for tsetse flies, and Zina Berman, Sonia Wang, Paul Graham, and Marek Chodakowski for their technical support. This work was supported by NSF Graduate Research Fellowships to JSS and NKL, by the Dwight N and Noyes D Clark Scholarship Fund to JSS, by a Scholar Award from International Chapter of the PEO. Sisterhood (JSS), by an NIH NRSA to JSC, by NIH T32 GM007499, by NIH U01 AI15648-02, and by NIH RO1 grants to JRC.

## Additional information

### Funding

| Funder | Grant reference number | Author |
|---|---|---|
| Dwight N. and Noyes D. Clark Scholarship Fund | | Jennifer S Sun |
| P.E.O. Scholar Award | | Jennifer S Sun |
| National Science Foundation | Graduate Research Fellowship Program | Jennifer S Sun Nikki K Larter |
| National Institutes of Health | National Research Service Award | J Sebastian Chahda |
| National Institutes of Health | T32 GM007499 | Jennifer S Sun |
| National Institutes of Health | U01 AI15648-02 | John R Carlson |
| National Institutes of Health | RO1s | John R Carlson |

The funders had no role in study design, data collection and interpretation, or the decision to submit the work for publication.

### Author contributions

Jennifer S Sun, Conceptualization, Data curation, Formal analysis, Supervision, Funding acquisition, Investigation, Visualization, Methodology, Writing—original draft, Writing—review and editing; Nikki K Larter, Conceptualization, Funding acquisition, Methodology, Writing—review and editing; J Sebastian Chahda, Funding acquisition, Investigation, Methodology, Writing—review and editing; Douglas Rioux, Ankita Gumaste, Validation, Methodology, Writing—review and editing; John R Carlson, Conceptualization, Supervision, Funding acquisition, Visualization, Writing—original draft, Writing—review and editing

**Author ORCIDs**
Jennifer S Sun (iD) http://orcid.org/0000-0002-4274-0504
Nikki K Larter (iD) http://orcid.org/0000-0002-1938-1929
John R Carlson (iD) http://orcid.org/0000-0002-0244-5180

**Decision letter and Author response**
Decision letter https://doi.org/10.7554/eLife.39249.018
Author response https://doi.org/10.7554/eLife.39249.019

## Additional files

### Supplementary files

• Supplementary file 1. The *Obp59a* deletion. Cut sites were located 1-nt downstream (first large capital letter) of the 5' end of the coding region, and 71-nt downstream (last large capital letter) of the 3' end of the coding region, thereby removing essentially all of the coding sequence (blue letters) of *Obp59a*. Homology arms were used to replace the coding region with a *DsRed* marker.
DOI: https://doi.org/10.7554/eLife.39249.015

• Transparent reporting form
DOI: https://doi.org/10.7554/eLife.39249.016

### Data availability

All data generated or analysed during this study are included in the manuscript and supporting files.

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
