## [Decision Letter]

Thank you for submitting your article "Humidity response depends on the small soluble protein *Obp59a* in *Drosophila*" for consideration by *eLife*. Your article has been reviewed by three peer reviewers, and the evaluation has been overseen by Leslie Griffith as Reviewing Editor and K VijayRaghavan as the Senior Editor. The following individual involved in review of your submission has agreed to reveal her identity: Rachel Wilson (Reviewer #2).

The reviewers have discussed the reviews with one another and the Reviewing Editor has drafted this decision to help you prepare a revised submission.

Summary:

Both the function of olfactory binding proteins as well as the molecular basis for hygrosensation are still very much open questions. In this paper, the authors provide evidence that a soluble extracellular protein in the *Drosophila* antenna (called *Obp59*) is a key part of the molecular mechanisms of hygroreception in this organism, bringing these two areas together in an unexpected way. The data in general are high quality and clearly presented. The reviewers had several issues that if addressed will improve the paper.

Essential revisions:

1) Are moist and dry sensing neurons still present in the *Obp59a* mutants? IR93a/25a/40a/68a Gal4 expression is only shown for the WT animals. Either survival or non-survival is interesting, but knowing the answer would be valuable.

2) As the authors point out, "It is conceivable that *Obp59a* mutants contain a subtle defect in the structure or composition of the sensillum." In Figure 3D

the antennal segment from mutant animals looks much different from WT. Is this accurate? Is the sacculus still open to the environment? Do other responses that require the sacculus (see Ai et al., 2010) still occur? Examining some other sacculus-dependent behavior e.g. the response to propionic/acetic acids (which has been mapped to chamber 3 of the sacculus) in the mutant would address this important point, though a higher resolution look at the sacculus would also help.

3) What change is occurring in these mutants that is endowing them with greater dehydration tolerance? To assess whether these changes directly reflect the loss of hygrosensing, it would be useful to test the desiccation tolerance of *Ir93a* mutants, which lack both moist and dry sensing. For *Obp59a*, are the mutant animals the same size as the WT? Other possibilities include that their surface area/volume ratios are different which could easily influence hygrosensory behaviors and/or desiccation resistance. Alternatively, the mutants might just spend less time searching for water (because they don't know they're dry), so they don't exhaust themselves in (hopeless) searching. If authors already have locomotor data already they should report it.

4) The results presented differ from published papers from other groups in several places. Authors show that flies prefer 96% to 70% relative humidity. But Perttunen and Salmi, 1956, Enjin et al., 2016 and Knecht et al., 2017 all reported that *Drosophila melanogaster* prefer 70% to higher levels. And in the experiments presented here, there is relatively little change in behavior after desiccation. Perttunen and Salmi, 1956, Enjin et al., 2016 and Knecht et al., 2017 all observed big switches in preference toward high humidity post desiccation. Potential reasons for these discrepancies in behavior should be discussed in text.

---

## [Author Response]

Essential revisions:1) Are moist and dry sensing neurons still present in the Obp59a mutants? IR93a/25a/40a/68a Gal4 expression is only shown for the WT animals. Either survival or non-survival is interesting, but knowing the answer would be valuable.

To address this question we have now carried out labeling experiments with three independent reagents: an anti-IR93a antibody, an anti-IR25a antibody, and an *IR68a-GAL4* driver. All three reagents labeled neurons in the second chamber of the sacculus of *Obp59a^1^,* in a pattern comparable to that observed in controls. These results have been added to the manuscript in a new figure, Figure 2—figure supplement 2.

2) As the authors point out, "It is conceivable that Obp59a mutants contain a subtle defect in the structure or composition of the sensillum." In Figure 3Dthe antennal segment from mutant animals looks much different from WT. Is this accurate? Is the sacculus still open to the environment? Do other responses that require the sacculus (see Ai et al., 2010) still occur? Examining some other sacculus-dependent behavior e.g. the response to propionic/acetic acids (which has been mapped to chamber 3 of the sacculus) in the mutant would address this important point, though a higher resolution look at the sacculus would also help.

We have now added to the manuscript new data that address, in both of the ways suggested by the reviewers, the possibility of a defect in the structure of the sacculus:

A) We used confocal microcopy to visualize at higher resolution the sacculus of both *Obp59a* and control, and found no gross defects in the morphology of the mutant. In particular, the mutant sacculus is still open to the environment. We have added a figure that illustrates these points (Figure 3—figure supplement 1). We have also added an acknowledgment that we cannot exclude the possibility of subtle morphological defects in sensilla.

B) As suggested, we tested the behavioral response of *Obp59a* mutant flies to both propionic and acetic acid and found no defects. We have added these data to Figure 5H.

3) What change is occurring in these mutants that is endowing them with greater dehydration tolerance? To assess whether these changes directly reflect the loss of hygrosensing, it would be useful to test the desiccation tolerance of Ir93a mutants, which lack both moist and dry sensing. For Obp59a, are the mutant animals the same size as the WT? Other possibilities include that their surface area/volume ratios are different which could easily influence hygrosensory behaviors and/or desiccation resistance. Alternatively, the mutants might just spend less time searching for water (because they don't know they're dry), so they don't exhaust themselves in (hopeless) searching. If authors already have locomotor data already they should report it.

This is an intriguing question, and we have explored it in several ways. As for size, we had considered the possibility that the greater desiccation resistance of *Obp59a* mutants might be due to smaller size and correspondingly greater surface area-to-volume ratio. We found that *Obp59a* females do in fact weigh less than controls, but that mutant males weigh the same as controls. Since both sexes show a desiccation resistance phenotype, but only females show the weight phenotype, the resistance seems not to be explicable simply in terms of weight. We prefer therefore not to include these data in the manuscript.

As for locomotor behavior, the revised manuscript shows that *Obp59a^1^* performs normally in a test of climbing behavior (Figure 5G).

4) The results presented differ from published papers from other groups in several places. Authors show that flies prefer 96% to 70% relative humidity. But Perttunen and Salmi, 1956, Enjin et al., 2016 and Knecht et al., 2017 all reported that Drosophila melanogaster prefer 70% to higher levels.

We have added a statement indicating that although the preference we observe in control flies between 96% and 70% RH is weak, the valence is opposite that found in these other studies. Knecht et al.. 2017, have shown that the valence of this decision is dependent on the hydration status of the flies. We have added the suggestion that the preference we have observed in our experiments may reflect the conditions in which we have cultured them.

And in the experiments presented here, there is relatively little change in behavior after desiccation. Perttunen and Salmi, 1956, Enjin et al., 2016 and Knecht et al., 2017 all observed big switches in preference toward high humidity post desiccation. Potential reasons for these discrepancies in behavior should be discussed in text.

We have added a statement indicating that the reason we did not observe the switch in preference demonstrated by Knecht et al., 2017, may have to do with differences in the procedures followed. We tested the effects of desiccation only in the faster hygrotaxis paradigm (Figure 4), which has a different duration, size, and geometry from that of Knecht et al., 2017. Also, the procedures used to desiccate flies are different: we did not provide the flies with a sucrose source during desiccation, which could affect their thirst level. N.B. We did not find a report of a change in preference in Enjin et al., 2016.)